# What Can We Learn from FGF-2 Isoform-Specific Mouse Mutants? Differential Insights into FGF-2 Physiology In Vivo

**DOI:** 10.3390/ijms22010390

**Published:** 2020-12-31

**Authors:** Friederike Freiin von Hövel, Ekaterini Kefalakes, Claudia Grothe

**Affiliations:** 1Institute of Neuroanatomy and Cell Biology, Hannover Medical School, Carl-Neuberg-Straße 1, D-30625 Hannover, Germany; Friederike.v.hoevel@web.de; 2Center for Systems Neuroscience (ZSN), University of Veterinary Medicine, Bünteweg 2, D-30559 Hannover, Germany; Kefalakes.Ekaterini@mh-hannover.de

**Keywords:** FGF-2ko, FGF-2LMWko, FGF-2HMWko, FGF-2LMWtg, FGF-2HMWtg, mu-tant mice, cardiovascular system, central nervous system, bone physiology

## Abstract

Fibroblast growth factor 2 (FGF-2), ubiquitously expressed in humans and mice, is functionally involved in cell growth, migration and maturation in vitro and in vivo. Based on the same mRNA, an 18-kilo Dalton (kDa) FGF-2 isoform named FGF-2 low molecular weight (FGF-2LMW) isoform is translated in humans and rodents. Additionally, two larger isoforms weighing 21 and 22 kDa also exist, summarized as the FGF-2 high molecular weight (FGF-2HMW) isoform. Meanwhile, the human FGF-2HMW comprises a 22, 23, 24 and 34 kDa protein. Independent studies verified a specific intracellular localization, mode of action and tissue-specific spatiotemporal expression of the FGF-2 isoforms, increasing the complexity of their physiological and pathophysiological roles. In order to analyze their spectrum of effects, FGF-2LMW knock out (ko) and FGF-2HMWko mice have been generated, as well as mice specifically overexpressing either FGF-2LMW or FGF-2HMW. So far, the development and functionality of the cardiovascular system, bone formation and regeneration as well as their impact on the central nervous system including disease models of neurodegeneration, have been examined. This review provides a summary of the studies characterizing the in vivo effects modulated by the FGF-2 isoforms and, thus, offers a comprehensive overview of its actions in the aforementioned organ systems.

## 1. Introduction

Fibroblast growth factor 2 (FGF-2) belongs to the FGF superfamily comprising at least 22 members in humans (FGF-1-14; FGF-16-23) and rodents (FGF-1-18; FGF-20-23) [1]. First isolated from the bovine pituitary gland, it received its primary name due to its pH-dependent dissociation from heparin—hence, basic FGF [2,3,4]. FGF-2 is expressed in various mammalian cell types and tissues, bearing different physiological and pathophysiological functions. Thereby, FGF-2 mRNA expression is enhanced in the heart, aorta, lung, white adipose tissue, and both the male and female reproductive tract [5,6]. Furthermore, FGF-2 is widely used in stem cell research as an agent of self-renewal (proliferation) and differentiation in vitro. Cultured stem cells first undergo a period of self-renewal during which they maintain an undifferentiated homogenous state of pluripotency or multipotency. The addition of different factors can act as stimuli of differentiation and, thus, promote cell fate commitment to lineages of interest [7,8,9,10]. Although many other signals are also involved in these processes, it is implied that the concentration of FGF-2 plays a pivotal role regarding cell fate specification [11,12].

Interestingly, based on the same mRNA, different FGF-2 isoforms are expressed (Figure 1A). In humans, starting at an AUG codon, an 18-kilo Dalton (kDa) FGF-2 protein is translationally initiated, termed low molecular weight FGF-2 (FGF-2LMW). Meanwhile, translation from alternative CUG codons results in a 22, 23, 24, and 34 kDa product, summarized as high molecular weight FGF-2 (FGF-2HMW) [13,14,15]. In rodents, only three FGF-2 isoforms exist; whereby, the 18 kDa FGF-2LMW isoform can be distinguished from two FGF-2HMW (21 and 22 kDa) isoforms [16,17,18,19]. Furthermore, protein expression of all known isoforms was demonstrated to differ regarding the tissue and the developmental stage [20,21,22,23].

Noteworthy, FGF-2LMW is physiologically located in the cell nucleus, cytoplasm and extracellular space. Thus, it can also be secreted by auto-/paracrine mechanisms or in other words “shedded” [24,25,26]. This renders FGF-2LMW to function through the established FGF receptors 1-4 (FGFR1-4), as well as through the recently included FGFR-5, and to induce canonical downstream signal transduction or result in nuclear translocation [26,27,28,29]. Hence, the activation of different downstream pathways regulates cell proliferation, migration, viability and cell fate commitment in a ligand-, receptor- and concentration-dependent manner [30]. In contrast, the FGF-2HMW isoforms, which display NH_2_-terminal extensions of the FGF-2LMW core sequence, have an additional nuclear localization signal restricting their expression predominantly to the cell nucleus and the ribosome [31,32,33,34]. This results in an intracrine mode of action by targeting nuclear proteins and transcription factors [35]. For example, in vitro FGF-2HMW was demonstrated to exclusively promote FGF-2 gene expression by activating the integrative nuclear FGFR-1 signaling (INFS) pathway [36,37]. Besides their different localization, the protein amount and distinct spatiotemporal expression patterns of all isoforms as well as various modes of action suggest defined physiological responsibilities in vivo. With regard to that, pharmacological modification of FGF-2 isoforms could also bear different therapeutic potential in disease state.

Since gene targeting is an established method for studying gene functions in vivo, different total FGF-2 knock out (ko) mice strains have been developed [17,18,19,38]. Of these, only the Fgf2^tm1Doe^ is commercially available today [17,39,40,41], having been analyzed more closely demonstrating FGF-2 mediated effects in wound healing, angiogenesis and stenosis as well as bone regeneration, and development of the central nervous system (CNS) [17,39,40,41].

Nevertheless, these FGF-2ko mice were not valid to distinguish between the physiological effects of the FGF-2LMW and FGF-2HMW isoforms. On this account, isoform-specific ko mice were developed: Mice lacking FGF-2LMW (Fgf2^tm2Doe^), and therefore only possessing the 21 and 22 kDa FGF-2HMW (below abbreviated FGF-2LMWko) [42], and vice versa—those lacking FGF-2HMW (Fgf2^tm3Doe^) and therefore only expressing the 18 kDa FGF-2LMW (below abbreviated to FGF-2HMWko) [43] (Figure 1B). Moreover, different transgenic (tg) mice overexpressing either the FGF-2LMW (below abbreviated to FGF-2LMWtg) or FGF-2HMW (below abbreviated to FGF-2HMWtg) isoform, in addition to endogenous FGF-2, have also been generated (Figure 1C). Thereby, the applied promoters differed with respect to the tissue of interest, to determine the physiological as well as pathophysiological impact of FGF-2LMW and FGF-2HMW in vivo. In detail, created FGF-2HMWtg mice either overexpress the human 24 kDa FGF-2 isoform driven by the protein kinase C (PGK) promoter [44] or combined the human 22, 23, and 24 kDa FGF-2 isoform using the collagen 3.6 (Col3.6) promoter [45,46]. The latter promoter was also used to generate FGF-2LMWtg mice overexpressing the human FGF-2LMW isoform as analyzed in detail with regard to bone pathophysiology [46,47]. In contrast, FGF-2LMWtg mice overexpressing the rat FGF-2LMW protein under the Rous sarcoma virus long terminal repeat (RSV) promoter were applied in cardiovascular research [48].

Within this review, we focused on results of studies using both, ko and tg FGF-2 isoform-specific mice, respectively, to provide a thorough characterization of the FGF-2 isoform mediated effects. This allows to more precisely determine the impact of a loss or gain of function of either FGF-2LMW or FGF-2HMW. Since the FGF-2 isoforms are interrelated with regard to their mode of action (for example, ko of one protein implies expression and action of the other), a clear separation by isoforms is impossible. Therefore, this review was structured according to insights gained from in vivo studies using FGF-2LMW and FGF-2HMW ko and tg mice regarding the development and functionality of the cardiovascular system, bone formation and regeneration, as well as central nervous system development and effects in neurodegenerative disease models.

## 2. FGF-2 Isoforms in the Cardiovascular System

Through development, FGF-2 is physiologically expressed in the nucleus and the cytoplasm of the heart mesenchyme, but only marginally detectable in the valvular endothelium. Furthermore, it can be found in the heart epicardium and myocardium, coronary vessels and smooth muscle cells of the aorta [49]. Interestingly, total FGF-2ko mice do not develop any morphological changes in the vascular or heart structure [17]. However, FGF-2ko mice show reduced vascular tone and lower arterial blood pressure, suggesting autonomic dysfunctions as examined in two independently generated FGF2ko strains [17,18]. Based thereon, FGF-2 isoform-specific ko mice have been characterized to more precisely determine the FGF-2 mediated effects in the cardiovascular system (Table 1). An extensive study by Nusayr and Doetschman [50], distinguishing not only between FGF-2 isoforms but also sex, revealed that although female FGF-2LMWko mice show increased body weights, they tend to have smaller hearts characterized by a reduced left ventricle (LV) mass and wall thickness (higher LV volume and internal dimension) which is indicative of a dilative cardiomyopathy. Regarding mitral valve flow, female FGF-2LMWko mice also displayed myocardial stiffness due to a higher early wave to atrial wave ratio (E/A ratio) combined with a restrictive diastolic filling pattern. A follow-up study also confirmed increased expression of atrial natriuretic factor (ANF) in these mice. In comparison to female FGF-2LMWko hearts, although male FGF-2LMWko mice have normal body- and heart weights (and LV heart volume), their hearts showed increased left atrial dimensions, a lower E/A ratio and a decline in LV pressure resulting in an overall impaired relaxation filling [50]. Apart from that, male FGF-2LMWko mice also showed a higher α-smooth muscle actin (α-SMA) expression [51]. Since alterations in heart elasticity often correlate with cardiomyopathies, FGF-2LMW seems to play a pivotal role in systolic and diastolic function and flow during heart development in both sexes, with a higher severity towards female mice when knocked out. This hypothesis was supported by the cardiac parameters of female FGF-2HMWko mice collected within the same study, which equaled those of wildtype (wt) littermates. This was also true for male FGF-2HMWko mice, which developed normal body and LV heart weights [50,51]. Notably, FGF-2HMWko mice rather had a tendency to lower body weights, while this was enhanced in FGF-2LMWko, even though this never reached significance compared to wt littermates [50]. Moreover, male FGF-2HMWko hearts showed a lower LV volume and internal dimensions in systole. This correlates with an increased systolic function as characterized by a higher cardiac output, stroke volume and fractional shortening. In addition, systolic function was also enhanced due to the higher velocity of both the pulmonary artery flow and the mitral valve flow as well as the pressure gradient of the mitral valve flow [50]. Finally, male FGF-2HMWko mice showed an increase in ANF expression [51]. Hence, loss of FGF-2LMW seems to be detrimental for proper cardiac function of both sexes, whereas FGF-2HMW seems to only affect male mice [50]. The same studies, however, did not show any isoform- or sex-dependent heart rate dysfunctions. Regarding cardiac development, no other studies analyzed the differences between FGF-2 isoforms and sex. Nonetheless, it has been previously demonstrated that the FGFR-1 switches from its long isoform to its short one during maturation of mouse hearts [52]. This switch was associated with reduced proliferation of adult cardiomyocytes upon FGF-2LMW stimulation, implying FGF-2HMW to be a regulator of proliferation of embryonic cardiomyocytes [53]. Indeed, proliferation was linked to FGF-2HMW in a study employing vascular smooth muscle cells obtained from the aortae of mice overexpressing human FGF-2LMW or FGF-2HMW. Within this study, overexpression of FGF-2HMW prominently stimulated cell proliferation by increased DNA synthesis upon additional serum treatment [46]. Therefore, it can be suggested that during embryonic stages of heart development, FGF-2HMW is crucial in retaining cardiac cells in growth, whereas once gender specification has taken place, both isoforms influence heart structure and function differentially through maturation in a sex dependent manner.

FGF-2 is a potent angiogenic factor with wound healing properties due to its ability to stimulate cell proliferation and differentiation [54]. Nevertheless, first characterizations of isoform-specific FGF-2LMWko and FGF-2HMWko mice revealed no obvious defects through vasculogenesis or angiogenesis compared to wt littermates [43,44,45,46,47,48,49,50,51,52,53,54,55]. However, a single study demonstrated that FGF-2HMW is the determining factor for the angiogenic effects mediated by 17 β -estradiol (E2) in female FGF-2LMWko mice [42]. This is supported by in vitro experiments where E2 treatment increased FGF-2HMW gene expression through FGFR-1 and FGF-2 interacting factor (FIF) in endothelial cells, resulting in increased cell migration [42]. In line with this enhanced expression of the heat shock protein 27 (HSP27) and simultaneous treatment with E2 resulted in the secretion of FGF-2HMW in endothelial cells [56]. Apart from its proliferative effects in endothelial cells, administering recombinant or endogenously produced FGF-2HMW was able to inhibit migration in vitro. Thereby, inhibition of migration could be blocked by antibodies specific against the N-terminal extension of FGF-2HMW. Since this is a prolongation of the FGF-2LMW core sequence shared by all FGF.2 isoforms, the effect was proven to exclusively depend on FGF-2HMW expression [57]. Although there seems to be a relation between E2 and FGF-2HMW, till today, comparable studies characterizing FGF-2HMWko female mice are still elusive, as is the interaction of testosterone with the expression of FGF-2 which has so far only been characterized in transgenic mouse models generated by microinjection of a plasmid construct expressing an FGF-2-IRES sequence [58].

Notably, both E2 and FGF-2 have been related to atherosclerotic pathophysiology, a chronic inflammatory syndrome affecting the arteries, which is linked to coronary heart disease [59,60]. In the established atherosclerotic apolipoprotein E (ApoE) ko mouse model, where mice developing atherosclerotic lesions following a high-fat diet, double ko mice lacking both ApoE and FGF-2LMW demonstrated that loss of FGF-2LMW reduces atherosclerotic aortic lesions in adult male mice. These mice also showed a decreased inflammatory response in terms of macrophage infiltration, oxidative stress as well as chemokine and adhesion molecule expression (MCP-1, VCAM-1, Nox4, and p47phox) [60]. Besides FGF-2LMW, E2 has also been shown to decrease blood pressure and atherosclerosis in the ApoE mouse model after its exogenous administration [61]. In fact, this effect has been attributed to the estrogen receptor α (ERα), since ko of ERα in the ApoE mouse model was hypothesized to be atheroprotective [61]. Notably, FGF-2HMW showed an estrogenic dependency regarding its fusion to the hormone-binding domain of the ER [62]. However, the extent of the hormone dependency of both FGF-2 isoforms upon the cardioprotective potential of estrogen and testosterone underlies future research, which should not only be isoform oriented but also consider sex, receptors and the separate processes of cardiac disease and remodeling.

Until now, the cardioprotective potential of the FGF-2 isoforms has been evaluated in ischemia-reperfusion (I-R) injury models as well as doxorubicin or isoproterenol treatment [44,49,51,55,63] (Appendix A). It has been shown that during I-R injury, lack of FGF-2LMW delays postischemic recovery, whereas loss of FGF-2HMW benefits both contractile and diastolic function [44,49,55]. In line with this, FGF-2LMWko mice have been demonstrated to perform similarly compared to FGF-2ko mice, leading to the conclusion that the loss of FGF-2LMW is detrimental for the FGF-2 cardioprotective potential [55]. During I-R injury, cardiotoxicity correlated with c-Jun N-terminal kinase (JNK) and mitogen activated kinase 4 and 7 (MKK4/7) activation. Specifically, MKK7/JNK pathway activation led to enhanced apoptosis as registered by activation of c-Jun as well as an increased caspase 3 cleavage and TdT-mediated dUTP-biotin nick end labeling (TUNEL) positive cells [55]. Furthermore, the authors suggested that released FGF-2LMW during I-R injury protects the heart through FGFR-1-mediated signal transduction [44,55]. Since FGFR-1 interacts with protein kinase C (PKC), a follow-up study also analyzed the cardioprotective potential of the LMW-PKC axis during I-R injury, but in FGF-2HMWko mice. Within this study, loss of FGF-2HMW benefited the contractile function after I-R injury by showing an enhanced activation of PKCα during ischemic onset and an increased activation of PKCε during the early reperfusion state. In addition, alterations in the localization of the myofilament PKC isoforms were registered, together with an increased activation of troponin I and T as well as MgATPase activity and a decreased calcium sensitivity. Generating FGF-2HMWko and PKCα double ko mice diminished this effect and resulted in decreased cardioprotection following I-R injury, which was accompanied by increased systolic function and a minor increase in PKCε expression. The detected decreased cardioprotection could only be attenuated by inhibition of PKCε [64]. Increased myocardial cell death present in those mice, led to the conclusion that FGF-2LMW was detrimental for the proper cardiac contraction following I-R injury, and that this cardioprotection was strongly PKCα dependent [44,55,64]. The cardioprotective potential of FGF-2LMW after I-R injury has not only been proved in FGF-2 isoform-specific ko mice but also in FGF-2 isoform overexpression models. Surprisingly, overexpression of the human 24 kDa FGF-2HMW in mice did not show differences in cardiac growth, vasculogenesis or angiogenesis. However, after I-R injury, ex vivo heart recovery was lowered with respect to contractile and relaxation function. These mice were also characterized by a reduced activation of the FGFR-1 [44]. Conversely, FGF-2LMW overexpression was shown to increase capillary density (in an FGFR-1 independent manner) concomitant with an activation of FGFR-1 downstream signal effectors including c-Jun, p38MAPK, PKCα and PKCε. Under I-R stress, isolated hearts received from those mice showed an increased myocyte viability as registered by lower lactate dehydrogenase levels [48]. Unlike FGFR-1, overexpression of FGFR-2 has been reported to benefit cardioprotection in myocardial ischemia in vivo. More valuably, improved infarct size, vessel formation and blood perfusion in these animals were registered, together with an increased gene expression of FGF-2. Furthermore, activation of the Akt pathway enhanced viability of cardiomyocytes and smooth muscle cells [65]. Since in vivo FGF-2HMW is restricted to the cell nucleus, it was hypothesized that within this study, increased FGF-2 mRNA correlated with higher levels of released FGF-2LMW protein, which induced a positive feedback loop of Akt activation, further enhanced survival of cardiomyocytes and smooth muscle cells. Altogether, these results indicate that the cardioprotective potential of FGF-2LMW strongly depends on the activated FGFR isoform and its specific downstream signal transduction, revealing different signaling effectors upon ischemic insult. In another study, the cardioprotective potential of both FGF-2 isoforms was evaluated by echocardiography following doxorubicin injection. In wt mice, doxorubicin normally results in a decreased systolic heart function and a dilative cardiomyopathy. However, such effects were not registered either in male or in female FGF-2HMWko mice, an outcome also confirmed by in vitro experiments using mouse embryonic fibroblasts [63]. In a pathologic cardiac hypertrophy and fibrosis model caused by injection of isoproterenol, a β-adrenergic agonist, both, female and male FGF-2LMWko and FGF-2HMWko mice showed hormone-dependent differences that were distinguishable for hypertrophy and fibrosis [51]. Interestingly, in this study, as already stated above, female FGF-2LMWko mice had smaller hearts (heart to body weight ratio) compared to their male and female wt and FGF-2HMWko littermates, resulting in cardiac hypoplasia. Moreover, herein, female FGF-2HMWko mice had smaller cardiomyocytes compared to male littermates, which is probably explained by reduced proliferation due to the lost interaction of E2 and FGF-2HMW [42]. However, following isoproterenol injection, cardiac hypertrophy was triggered in all groups, except for female FGF-2HMWko mice [51]. Thereby, it was mostly enhanced in male FGF-2HMWko mice. Meanwhile, female and male FGF-2LMWko mice reacted in a comparable manner to their wt littermates, independent of sex. Notably, fibrosis in female mice was similar to wt littermates regardless of FGF-2 isoform. The same was true for male FGF-2HMWko mice, whereas male FGF-2LMWko animals developed excessive fibrosis. The male FGF-2LMWko mice also displayed the highest increase in α-SMA and collagen I expression following isoproterenol injection. Meanwhile, in FGF-2HMWko littermates, an induction of ANF was reported [51]. Overall, FGF-2 has been reported to stimulate proliferation of both fibroblasts and cardiomyocytes (hypertrophy and fibrosis), which, in turn, results in tissue scarring upon cardiac insult [66,67]. Importantly, it was shown that both hypertrophy and fibrosis undergo a sex- and isoform-specific regulation after isoproterenol treatment. Interestingly, a previous study demonstrated that adding FGF-2LMW to neonatal myocytes induced reprograming of the cells to a “fetal” phenotype [68]. On the contrary, the same mechanism was suggested for FGF-2HMW after adding isoproterenol. In this study, a time-dependent shift between isoform activation was observed, hypothesizing that both FGF-2 isoforms play different roles regarding the short- and long-term responses to isoproterenol treatment. FGF-2HMW, which sustains the proliferative state of immature hearts, was shown to be highly activated 24 h after isoproterenol treatment, thus stimulating the regeneration as part of the repair response. In comparison, FGF-2LMW showed a sustained activation throughout different time-points [69].

In conclusion, since the differences in gene expression, protein translation, activation and release of FGF-2 isoforms in the cardiovascular system vary depending on sex, cell type, tissue damage as well as the time point of the analysis, it is a prerequisite to consider each of these factors when examining the FGF-2 mediated effects. 

**Table 1 ijms-22-00390-t001:** Characterization of the cardiovascular system of male (♂) and female (♀) FGF-2 isoform-specific mouse mutants through development. Alterations were provided as increased (↑) or decreased (↓) compared to wildtype littermates for either FGF-2 isoform-specific ko mice (FGF-2LMWko and FGF-2HMWko), or mice additionally overexpressing rat FGF-2LMW (FGF-2LMWtg) or human 24 kDa FGF-2HMW (FGF-2HMWtg). To increase understandability, findings were separated by sex whenever possible and mentioned at their first description. The FGF-2 isoform-specific cardioprotective potential was evaluated in different disease models summarized in Appendix A. α-SMA, α-smooth-muscle actin; ANF, atrial natriuretic factor; ApoE, apolipoprotein E; E/A ratio, early wave to atrial wave ratio; FGF-2, fibroblast growth factor 2; HMW, high molecular weight; JNK, c-Jun N-terminal kinase; ko, knock out; LWW, low molecular weight; LV, left ventricular; MAPK, mitogen activated protein kinase; MCP-1, monocyte chemoattractant protein-1; Nox4, NADPH oxidase 4; PGK, phosphoglycerate kinase; PKC, protein kinase C; RSV, Rous sarcoma virus long terminal repeat; tg, transgene; VCAM-1, vascular cell adhesion protein 1; ^§^ compared to FGF-2ko.

Strain	FGF-2 LMWko (FGF-2^tm2Doe^)	FGF-2 HMWko (FGF-2^tm3Doe^)	FGF-2HMWtg Overexpressed Human 24 kDa Driven by PGK Promoter	FGF-2LMWtg Overexpressed Rat 18 kDa Driven by RSV Promoter
Phenotype	♂ ↑ left atrial dimensions ↓ E/A ratio ↓ decline in left ventricular pressure ↑ isovolumic relaxation time (impaired elaxation filling patterns) (Nusayr and Doetschman 2013) [50] ↑ α-SMA expression (Nusayr, Sadideen et al. 2013) [51] ♀ ↑ body weight ↑ heart hypoplasia ↓ LV posterior mass and wall thickness in iastole and systole ↑ LV volume in systole and diastole ↑ LV internal dimension in diastole ↑ E/A ratio ↑ myocardial stiffnessrestrictive diastolic filling patterns (Nusayr and Doetschman 2013) [50] ♀ ↑ ANF expression (Nusayr, Sadideen et al. 2013) [51] ♀ ↑ angiogenesis mediated by 17β-estradiol ^§^ (Garmy-Susini, Delmas et al. 2004) [42]	♂/♀ ↑ FGF-2LMW protein levels in heart tissue, capillaries and vessels (Azhar, Yin et al. 2009) [43], (Liao, Bodmer et al. 2010) [44] ♂ ↓ LV volume in systole ↓ LV internal dimensions in systole ↑ systolic function indicated by greater cardiac output, stroke volume and fractional shortening ↑ flow velocity in the pulmonary artery ↑ mitral valve flow mean velocity and mitral valve flow mean pressure gradient (Nusayr and Doetschman 2013) [50] ♀ ↑ ANF expression (Nusayr, Sadideen et al. 2013) [51] ♀ no alterations compared to wt littermates (Nusayr and Doetschman 2013) [50] (Nusayr, Sadideen et al. 2013) [51]	♂/♀ no alteration regarding heart growth, vasculogenesis/ angiogenesis or endogenous FGF-2 protein expression (Liao, Bodmer et al. 2010) [44]	♂/♀ ↑ capillary density ↑ levels of phosphorylated c-Jun ↑ levels of phosphorylated p38MAPK ↑ levels of phosphorylated membrane associated PKCα and cytosolic PKCε (Sheikh, Sontag et al. 2001) [48]
Atherosclerosis	FGF-2^tm2Doe^ x ApoEko ♂ ↓ atherosclerotic lesions in the aorta ↓ macrophage infiltration ↓ MCP-1, VCAM-1, Nox4 and p47phox expression (Liang, Wang et al. 2018) [60]			

## 3. FGF-2 Isoforms in Bone Physiology

As previously demonstrated in physiological bone development FGF-2 mRNA and protein expression were verified in periosteal cells, osteoblasts, osteocytes and chondrocytes [5,70,71]. Even though FGF-2ko mice were morphologically normal, they developed reduced bone mass characterized by significantly less bone volume density and a lower trabecular number, accompanied by increased trabecular separation through aging [20,72]. Moreover, the loss of FGF-2 was accountable for the reduced number of osteoclasts, altered proliferation and differentiation of osteoblasts, and development of osteoarthritis (OA) and osteoporosis compared to wt littermates [18,71]. Interestingly, similar but worse morphometric alterations have been described for mice, overexpressing all human FGF-2 isoforms, ultimately leading to the expression of a dwarfed phenotype [20,73].

Based thereon, FGF-2 isoform-specific ko mice have been characterized to more precisely determine the specific role of the FGF-2 isoforms in bone physiology and pathophysiology. In addition to that, transgenic mice overexpressing either human 18 kDa FGF-2LMW (FGF-2LMWtg) or 22, 23 and 24 kDa FGF-2HMW (FGF-2HMWtg) with high specificity in osteoblast lineage cells but also osteoclasts have been generated [45,47,74,75]. Since the importance of FGF-2 and its mode of action in bone pathophysiology especially related to human disease had been previously discussed by Coffin, et al. [76], within this review we aimed to provide the main in vivo findings related to the mentioned strains. Thereby, the characteristic phenotype developed through maturation (Table 2) of each strain is described first, followed by results of specific research topics including ageing/osteoarthritis (Table 2), fracture healing, phosphate diet as well as FGF-23 and FGFR-1 blockade (Appendix A). Since we aimed to focus on in vivo outcomes, results obtained by in vitro experiments within the same studies are discussed as supporting evidence to the animal studies and are listed as a separate table in the Appendix A.

Similar to FGF-2ko mice, adult male FGF-2LMWko animals developed morphometric reduced bone mineral content and density without apparent phenotype [47]. At the same time, increased expression of the secreted frizzled related protein 1 (sFRP1) was observed in the bones of these mice. sFRP1 is a known antagonist of the Wnt/β-catenin pathway, negatively influencing bone formation by downregulation of genes responsible for collagen synthesis (e.g., type I collagen (*Col1*)) or osteoblast differentiation (e.g., *runt-related transcription factor 2* (*Runx2*), *osterix*, *osteocalcin* (*Oc*), *osteopontin* (*Op*)). Furthermore, in vitro depletion of sFRP1 in bone marrow stroma cells of FGF-2LMWko mice partially reverse this effect [47], suggesting that FGF-2LMW had prominent anabolic influence on osteoblasts. This hypothesis was further supported by the immediate comparison to male FGF-2LMWtg mice, which developed increased bone mineral content and density, coincident with reduced sFRP1 and increased β-catenin expression [47,77].

In accordance with that, FGF-2HMWko mice also display increased bone mass in trabecular and cortical bone, depicted by increased femoral and vertebral bone volume density, trabecular number and thickness, while reduced trabecular spacing and cortical porosity, as well as increased bone mineralization, formation and connective tissue density [78]. Besides the upregulation of the aforementioned genes (*Col1*, *Runx2*, *osterix*, *Oc* and *Op*) through Wnt signaling, sclerostin (*Sost*), another Wnt inhibitor gene, and its associated protein sclerostin, hampering osteogenesis, were reduced in osteocytes of these mice [78]. Additionally, pro-mineralization genes like *dentin-matrix phosphoprotein 1* (*Dmp1*) belonging to the small integrin-binding ligand N-linked glycoprotein (SIBLING) family, and its interaction *phosphate-regulating neutral endopeptidase X-linked gene* (*Phex*) were also increased in bone documenting exaggerated osteoblast function [78]. Consequentially, reduced *Fgf-23* mRNA expression was detected in bone compared to wt littermates, influenced by *Phex* and *Dmp1* through FGFR-1, which, in turn, was upregulated [79,80]. Meanwhile, serum levels of FGF-23 were not changed compared to wt, which was also true for phosphate, calcium, and parathyroid hormone (PTH) levels in FGF-2HMWko, as well as FGF-2LMWtg mice [47,78].

FGF-2HMWtg mice developed the most severe dysgenesis phenocopying the Hyp mouse, an established model for human X-linked hypophosphatemic rickets [45,81]. Independent of sex, those mice analyzed developed dwarfism, along with reduced bone mineral density and content and also rickets/osteomalacia [45,79,82,83]. Histomorphometric investigations supported this phenotype, demonstrating predominant bone resorption processes, while a decreased bone formation was registered [45]. In line with this, serum PTH levels were also elevated. Appropriate gene analysis verified a reduced *Col1* and *Oc* expression, while increased *Op* and *matrix γ-carboxyglutamic acid protein (Mgp)* expression were evident, which is a mineralization inhibitor [45,84]. Furthermore, Fgf-23 mRNA and protein levels were also enhanced in osteoblasts and osteocytes, and highly elevated in serum of FGF-2HMWtg mice [45,82,83]. Along with this, these mice developed hypophosphatemia due to renal phosphate wasting, resulting in phospaturia [45,82,83,84]. Meanwhile, hematocrit, serum creatinine, calcium and 1,25-dihydroxy-vitamin D (1,25D) levels were comparable to wt littermates [45]. Taken together, besides elevated PTH concentrations, increased FGF-23 in FGF-2HMWtg mice activates the renal Klotho-FGFR-1-pErk1/2 complex, resulting in downregulation of the type 2a sodiumphosphate co-transporter (NPT2a) in the proximal tubulus of the kidney, additionally leading to less phosphate absorption [45,82,84,85]. In order to counteract hypophosphatemia in FGF-2HMWtg mice, a promising approach was achieved by a continuous high phosphate diet, which was able to normalize the exaggerated serum FGF-23 level [45]. Unfortunately, this diet alone was not sufficient to alleviate all described symptoms. However, it might act synergistically to FGF-23 neutralizing antibody treatment, which improved phosphate homeostasis by downregulation of the FGF-23 renal co-receptor *Fgfr-3c/Fgfr-1* and *klotho*, resulting in upregulation of renal *NPT2a* following single injection in male FGF-2HMWtg mice [83]. Meanwhile, serum PTH and 1,25D were not affected by this treatment, which probably was mediated through simultaneous upregulation of both 1,25D regulating genes, renal *25-hydroxyvitamin D 1 α-hydroxylase (CYP27B1)* and renal *25-hydroxyvitamin D 24-hydroxylase (CYP24)*. These results were mainly confirmed by a study in female FGF-2HMWtg mice comparing single or long-term treatment of FGF-23 neutralizing antibody [79]. However, female FGF-2HMWtg had higher serum 1,25D concentrations, even though degrading *CYP24* expression was further increased. Additionally, long-term treatment not only rescued phosphate homeostasis but also resulted in morphometric improvements, demonstrated by increased vertebral bone mineral content and density as well as increased femoral trabecular number, while decreased intertrabecular spacing and cortical porosity were evident. Notable, not all mineralization defects were restored, even though also enhanced expression of the SIBLING family members, *matrix extracellular phosphoglycoprotein* (*Mepe*), *Op*, *Dmp4*, and *bone sialoprotein (Bsp)*, promoting bone mineralization were reported [79]. Therefore, female FGF-2HMWtg mice showed increased expression of genes (*Enpp1*, *Ank*, *Slc20a1*) responsible for the increasing extracellular concentrations of the inorganic pyrophosphate, which is a known inhibitor of matrix mineralization, while decreased expression of its degradation enzyme, nonspecific alkaline phosphatase (TNAP) was registered [79].

Besides targeting FGF-23, consequences following single application of the FGFR inhibitor NVP-BGJ398 in female FGF-2HMWtg mice have been analyzed [82]. This treatment also significantly influenced FGF-23/FGFR/Klotho/ERK signaling in the kidney, resulting not only in normalized serum and urinary phosphate levels but also a significantly exaggerated serum PTH, FGF-23 and 1,25D concentration [82]. Consequently, the FGFR inhibitor also decreased elevated renal protein expression of the Wnt inhibitors Sost and engrailed-1 (EN-1). A follow-up study, evaluating short- and long-term treatment with the same inhibitor in male FGF-2HMWtg mice, verified that a single dosage was adequate to improve hypophosphatemia and phosphaturia through NPT2a modulation [84]. Additionally, long-term treatment significantly increased mineralization and the bone formation rate equalizing whole body bone mineral content and density. Furthermore, femur, tibial and tail length were normalized after eight-week application period, significantly improving the dwarfed phenotype of FGF-2HMWtg mice. Thereby, similar to the FGF-23 neutralizing antibody, FGFR inhibition resulted in the upregulation of femoral *Fgfr-1c* as well as members of the SIBLING family. Notably, while cortical porosity was normalized and bone length was improved compared to wt femur, other parameters like trabecular number and thickness were not corrected. This probably could be a result of a still elevated serum FGF-23 level, following long-term FGFR inhibition [84]. Meanwhile, in vitro studies performed in chondrocytes and bone marrow derived stem cells (BMSCs) demonstrated increased FGF-23 gene or protein expression. This coincided with chondrocyte hypertrophy and increased expression of Wnt signal transducers as well as increased mineralization of bone nodule formation and osteogenic markers. Thereby, inhibition of either FGF-23, Wnt or FGFR-1 was able to protect this phenotype and ameliorate the pathologic hallmarks in cells received from FGF-2HMWtg mice [86,87]. Taken together, blocking FGF-23 and FGFR ameliorates some of the pathophysiologic effects in FGF-2HMWtg mice. Therefore, manipulation of the downstream-interrelated FGFR signal transduction effectors such as ERK1/2, AKT, Smad and β-catenin seems promising for further therapeutical interventions.

In addition to osteoporosis, FGF-2ko mice also tend to develop OA through aging (Table 2), which was more prominent following surgical induction [71,72,88]. Directly compared to that, FGF-2LMWko mice also develop OA in knee joints from six months onwards and independent of the sex analyzed [71]. Thereby, OA was characterized by tendonitis and arthritis, followed by loss of articular cartilage, flattening of the tibial plateau, osteophyte formation, and thinning of the femoral subchondral bone. In line with this, similar observations were described for male FGF-2HMWtg mice earlier [89]. Conversely, no radiographic changes have been reported in knee joints of male FGF-2LMWtg mice or male and female FGF-2HMWko mice through aging [71,89]. To summarize, this suggests a protective potential for FGF-2LMW against the development of OA and further underlines the catabolic influence of FGF-2HMW. This hypothesis was further supported by challenging FGF-2HMWko mice to cyclic tibial loading without induction of the OA phenotype, whereby FGF-2LMWko and FGF-2ko already developed first OA changes after two weeks, including signs of bone remodeling and cartilage damage [71]. Interestingly, compared to wt littermates, no alterations have been described in FGF-2HMWko mice except for a decreasing FGF-2LMW content, while increased activated FGFR-3 protein expression in articular cartilage through aging was apparent. On the other hand, FGF-2LMWko mice enhanced FGF-23 and FGFR-1 protein expression, leading to activation of ERK1/2. Interestingly, earlier ex vivo experiments using human articular chondrocytes verified that FGFR-3 signaling mediated anabolic gene expression, while FGFR-1 was the prominent isoform in OA preferentially activated by FGF-2, resulting in increased expression of the cartilage degrading enzymes *a disintegrin and metalloproteinase with thrombospondin motifs 5* (*ADAMTS-5*) and *matrix metalloproteinase 13* (*MMP13*) [90]. In line with this, *ADAMTS-5* and *MMP13* were enhanced in articular cartilage of FGF-2LMWko mice at the beginning of the obvious OA phenotype. Notably, prior to the appearance of these effects, upregulation of genes involved in the hypertrophic differentiation of chondrocytes, also encouraging OA development, was reported in young adult FGF-2LMWko mice [71]. This was also true for eight-week-old male FGF-2HMWtg mice compared to wt littermates [89]. For example, in FGF-2HMWtg mice, absence of the Wnt inhibitor *Sost* and downregulation of the *Dickkopf* Wnt signaling pathway inhibitor (*Dkk1*) gene was demonstrated [87]. At the same time, expression of phosphorylated *low-density lipoprotein receptor-related protein 5 (LRP5)*, *WNT5A* and *WNT7B* as well as phosphorylated inactive *glycogen synthase kinase 3β* (*GSK3β*) were greatly increased, which was further supported by in vitro experiments showing enhanced hypertrophic differentiation of chondrocytes from FGF-2HMWtg mice [87].

Since FGF-23 expression was also related to chondrocyte hypertrophy by upregulation of *Col10* and *Mmp13* in FGF-2HMWtg mice, the effects of long-term FGF-23 neutralizing antibody treatment were analyzed with respect to OA development [87,89,91]. Modulating the FGF-23/Wnt/β-catenin pathway partially improved the OA phenotype of adult female FGF-2HMWtg mice [87]. A follow-up study further demonstrated that long-term application of the FGFR tyrosine kinase inhibitor (NVP-BGJ398) also ameliorated the OA phenotype, including a reduction in bone morphometric changes and increased articular cartilage thickness and proteoglycan content in male, but not female, mice [92]. Thereby, decreased FGF-23 levels in serum and articular cartilage were measured, as well as reduced expression of genes involved in Wnt signaling, as confirmed earlier by in vitro observations [93].

Based on the anabolic effects mediated by FGF-2LMW, improved healing potential following bone defects has been hypothesized [47,77,94]. Following calvarial defects, it was confirmed that FGF-2LMW overexpression accelerated bone healing by promoting osteoblast activity due to the upregulation of *osterix*, *Oc*, and *Runx2* via Wnt/β-catenin signaling and simultaneous downregulation of *sFRP1* [47]. Furthermore, local addition of the BMP2 improved bone formation and mineral apposition rate, resulting in complete healing of calvarial defects [77]. Another study, evaluating closed tibial fracture healing in female FGF-2LMWtg mice fracture model supported the beneficial role of FGF-2LMW [94]. Within this study, the faster recovery included earlier callus development and bone remodeling through accelerated chondrocyte, and osteoblast precursor differentiation demonstrated by altered gene regulation [94].

Overall, the results of FGF-2 isoform-specific mice regarding bone physiology indicated catabolic effects mediated by FGF-2HMW isoforms, ultimately leading to a dwarfed phenotype as described for FGF-2HMWtg mice. In contrast, the FGF-2LMW isoform was associated with anabolic effects in bone development and also seemed to bear protective potential in bone pathophysiology. With regard to the observations in FGF-2tg and FGF-2HMWtg mice, which both also possess FGF-2LMW, it could be hypothesized that FGF-2HMW signaling overshadows FGF-2LMW function as long as it is expressed. Besides the specific cellular localization of each FGF-2 isoform, different FGFR affinity also played a role. Thereby, earlier studies already demonstrated that FGF-2HMW regulated other gene expression, including its own, through activation of nuclear FGFR-1, whereas FGF-2LMW did not [36]. This might explain why FGF-2HMWko mice show a decreased FGF-2 expression through aging [71], leading to less stimulation of the FGFR-1-Erk/MAPK pathway [90], but increasing content of FGFR-3, which was associated with protection against OA. On the other hand, OA developing FGF-2LMWko mice enhanced FGFR-1 expression and an elevated FGF-23 concentration mediated by FGF-2HMW [71,87]. Thereby, FGF-23 functions as an endocrine growth factor and mainly influences bone homeostasis through modulating the renal phosphate absorption, also via FGFR-1 signaling. Importantly, elevated serum FGF-23 levels, accompanied with hypophosphatemia, were associated with different syndromes in mice and humans [95], which is far beyond the aim of this review. On the contrary, this further highlights the high range of FGF-2 isoforms regarding bone physiology, which can not only be explained by one functional pathway. For this reason, FGF-2 isoform-specific mice display a unique tool for future investigations in bone pathophysiology, despite the difficulties in finding a single therapeutic target regarding the expression of FGF-2 isoforms. However, this opens the opportunity for multimodal therapy approaches, which might involve a combination of receptor inhibitor, specific antibody treatment and supplemental therapy.

**Table 2 ijms-22-00390-t002:** (**A**) Characterization of the bone related phenotype of adult FGF-2 isoform-specific mouse mutants in chronological order. Main findings were listed for either FGF-2 isoform-specific ko mice (FGF-2LMWko and FGF-2HMWko), or mice additionally overexpressing human FGF-2LMW (FGF-2LMWtg) or FGF-2HMW (FGF-2HMWtg). To increase understanding, findings were mentioned at their first description separated for male (♂) and female (♀) mice, whereby so far, only female FGF-2HMWtg mice have been analyzed more closely. Thereby, data were provided as increased (↑) or decreased (↓) compared to wt littermates. Moreover, data were not listened when confirmed in follow up studies in the respective literature provided below, but can completely be found in Appendix A. 1,25D, 1,25-dihydroxyvitamin D; ALP, alkaline phosphatase; Col1a1, Type I collagen; CTX, c-terminal telopeptide of type 1 collagen; Cyp24, renal 25-hydroxyvitamin D 24-hydroxylase; Cyp27b1, renal 25-hydroxyvitamin D 1alpha-hydroxylase; Dmp, Dentin matrix phosphoprotein; Egr-1, early growth response-1 transcription factor; En-1, Engrailed-1; Enpp1, Ectonucleotide pyrophosphatase/phosphodiesterase family member 1; FGF, fibroblast growth factor; FGFR, fibroblast growth factor receptor; Gsk3β, Glycogen Synthase Kinase 3 Beta; HMW, high molecular weight; knock out; LWW, low molecular weight; Mepe, Matrix extracellular phosphoglycoprotein; Mgp, Matrix gla protein; Npt2, sodium phosphate co transporter; Oc, Osteocalcin; Op, Osteopontin; Phex, Phosphate-regulating neutral endopeptidase; PTH, Parathyroid hormone; PTHR1, parathyroid hormone 1 receptor; Runx2, runt-related transcription factor 2; sFRP-1, secreted frizzled receptor 1; Slc20a1, Sodium-dependent phosphate transporter 1; Sost, sclerostin; Sostdc-1, Sclerostin domain-containing-1; tg, transgene; TNAP, tissue nonspecific alkaline phosphatase. (**B**) Characterization of the bone related phenotype developed through aging of FGF-2 isoform-specific male (♂) and female (♀) mouse mutants in chronological order. Alterations through aging were listened for either FGF-2 isoform-specific ko mice (FGF-2LMWko and FGF-2HMWko), or mice additionally overexpressing human FGF-2LMW (FGF-2LMWtg) or FGF-2HMW (FGF-2HMWtg). To increase understanding, findings were mentioned at their first description separated by sex whenever possible. Thereby, data were provided as increased (↑) or decreased (↓) compared to wt littermates. Moreover, data were not listed when confirmed in follow up studies in the respective literature provided below, but can be found in the Appendix A. Adamts5, A disintegrin metalloproteinase with thrombospondin motifs 5; Bax, B-cell lymphoma 2 associated X, apoptosis regulator; BMP, Bone morphogenetic protein; Col10, type 10 collagen; Dkk1, Dickkopf-Like Protein 1; FGF, fibroblast growth factor; FGFR, fibroblast growth factor receptor; Gsk3β, Glycogen Synthase Kinase 3 Beta; Hif1α, hypoxia inducible factor 1; HMW, high molecular weight; IL-1β, interleukin-1 β; Igf1, insulin like growth factor 1; ko, knock out; Lef1, Lymphoid Enhancer Binding Factor 1; LWW, low molecular weight; Lrp, low density lipoprotein receptor-related protein; Mmp, Matrix metallopeptidase; OA, Osteoarthritis; Sox9, Sex-determining region Y box 9; Sost, sclerostin; tg, transgene; Vegf, vascular endothelial growth factor; Wnt, wingless-type.

(A)				
Strain	FGF-2LMWko (FGF-2^tm2Doe^)	FGF-2HMWko (FGF-2^tm3Doe^)	FGF-2HMWtg Overexpressed Human 22, 23, 24 kDa Driven by Col3.6 Promoter	FGF-2LMWtg Overexpressed Human 18 kDa Driven by Col3.6 Promoter
Phenotype	♂↓ vertebral bone mineral density and content ↑ sFRP1 protein levels in trabecular bones (Xiao, Liu et al. 2009) [47]	♂↑ whole body bone mineral density and content ↑ vertebral, femoral bone mineral density and content ↑ femoral bone volume, trabecular thickness, number (cortical bone area, thickness, cortical mask) ↓ femoral trabecular spacing ↑ connective tissue density ↓ cortical porosity, bone resorption (↓ osteoclast surface, number) ↑ bone formation in cortical periosteum, trabecular bone (↑ osteoblast surface, inter-label thickness, mineral apposition rate) ↑ tibial Col1a1, Runx2, osterix, oc, op, Dmp1 gene expression ↓ femoral *Sost* gene expression ↓ serum sclerostin, protein levels ↓ tibial *Fgf-2, Fgf-23* gene expression (Homer-Bouthiette, Doetschman et al. 2014) [78]	♂dwarfism, osteomalacia ↓ body weight ↓ whole body bone mineral density and content ↓ femoral bone length ↓ vertebral volume, bone mineral density and content ↓ femoral bone volume, trabecular number, thickness ↑ femoral trabecular spacing ↑ bone resorption (↑ osteoclast surface, number) ↓ bone formation (↓ osteoblast, mineralization surface, bone formation rate) ↓ tibial *Col1a1, Oc* gene expression ↑ tibial *Op, Mgp* gene expression↓ serum phosphate ↑ serum PTH, CTX, FGF-23 ↑ tibial, femoral *Fgf-23, Phex* gene expression, protein levels ↑ renal *Fgfr-1c, Fgfr-3c, Klotho* gene expression ♂ with continuous phosphate diet ↑ body weight, bone mineral content and density ↑ serum phosphate to a normal level ↑ serum FGF-23 (Xiao, Naganawa et al. 2010) [45] ♂↓ tibia bone mineral density and content ↑ renal FGFR-1, FGFR-3, Klotho, C-Fos, activated ERK protein levels ↑ renal *C-fos, Egr1* gene expression ↓ renal *Npt2* gene expression ↑ renal *Cyp24, Cyp27b1* gene expression ↓ renal Npt2 protein levels (Du, Xiao et al. 2017) [83] ♂↓ tail length ↓ femoral length ↑ urinary phosphate level ↑ cortical porosity, trabecular spacing, osteoid volume ↓ cortical thickness, tissue ↓ endosteal/periosteal perimeter, subendosteal area ↓ mineralization of cortical bone area, metaphyseal cancellous bone volume, trabecular number ↑ femoral *Fgfr-3c, Pthr1, Op, Mgp* gene expression (Xiao, Du et al. 2017) [84]	♂↑ vertebral, tibial, femoral bone mineral density and content ↑ femoral bone volume, trabecular thickness, cortical bone area, thickness ↓ *sFrp1* gene expression, protein levels in trabecular bones ↑ *β-catenin* gene expression, protein levels (Xiao, Liu et al. 2009) [47] ♂↑ *Fgfr-1, Fgfr2, oc, β-catenin* gene expression in calvaria bone ↓ *sFrp1* gene expression in calvaria bone ↑ calvarial inter-label thickness, mineral apposition rate (Xiao, Ueno et al. 2014) [77]
Strain	FGF-2LMWko (FGF-2^tm2Doe^)	FGF-2HMWko (FGF-2^tm3Doe^)	FGF-2HMWtg Overexpressed human 22, 23, 24 kDa driven by Col3.6 promoter	FGF-2LMWtg Overexpressed human 18 kDa driven by Col3.6 promoter
Phenotype			♀↓ body weight ↓ femoral, tibial, vertebral bone mineral density and content ↓ serum phosphate ↑ serum FGF-23, 1,25D ↑ urinary phosphate level ↑ renal FGFR-1, En-1, klotho protein levels ↑ renal Klotho, Sostdc-1, En-1, Cyp24 gene expression ↑ activated renal ERK, Gsk-3β (Tr216) protein levels ↓ renal *Npt2, Akt* gene expression ↓ activated renal Gsk-3β (Ser9), active β-catenin and Akt protein levels (Du, Xiao et al. 2016) [82] ♀↓ femur length, cortical density, mineral apposition rate ↑ cortical porosity ↓ femoral bone volume, trabecular number ↑ femoral trabecular spacing ↑ osteoid volume ↑ serum ALP ↓ serum TNAP ↓ TNAP activity in osteocytes ↑ renal *Fgfr-1c, Fgfr-3* gene expression ↑ tibia Fgf-2, Fgfr-1c, Col1a1, Mgp, Dmp4, Phex, Mepe, Enpp1, SLc20a1 gene expression ↓ tibia *Dmp1, Rankl, Oc* gene expression ↑ femur cortical ERK, FGFR-1 protein levels (Xiao, Homer-Bouthiette et al. 2018) [79]	
**(B)**				
**Strain**	**FGF-2 LMWko** **(FGF-2^tm2Doe^**)	**FGF-2 HMWko** **(FGF-2^tm3Doe^**)	**FGF-2HMWtg** **Overexpressed Human 22, 23, 24 kDa Driven by Col3.6 Promoter**	**FGF-2LMWtg** **Overexpressed Human 18 kDa Driven by Col3.6 Promoter**
Aging/Osteo-arthritis	♂/♀ ↑ OA in knee joints (flattening of tibial plateau, osteophyte formation) ♂ ↓ femoral, tibial bone volume, trabecular number, thickness ↑ femoral, tibial trabecular spacing ↓ proteoglycan content, cartilage thickness in knee joint ↑ tendonitis, arthritis ↑ MMP-13, ADAMTS-5, FGF-2, FGF-23, FGFR-1 protein levels in articular cartilages ↑ Igf1, IL-1β, Bmp4, Hif1α, Bax, Fgf-2, Fgf-23, Fgfr-3 Vegf, Col10 gene expression in knee joints ↑ activated ERK protein levels in articular cartilage ↓ activated FGFR-3 in articular cartilage ↑ signs of OA following tibial loading (loss of proteoglycan content, thinning of subchondral bone) (Burt, Xiao et al. 2019) [71]	♂/♀ no radiographical signs of OA in knee joints ♂ ↑ activated FGFR-3 protein levels in knees ↓ FGF-2 protein levels in articular cartilage (Burt, Xiao et al. 2019) [71]	♂ ↑ OA in knee joints (flattening of tibial plateau, osteophyte formation, femoral subchondral bone thinning, sclerotic bone development, narrowing of the patellofemoral space, loss of trabeculae, sclerosis of femur) ↓ epiphyseal bone volume density, trabecular thickness, number in femur, tibiae ↓ proteoglycan content, cartilage thickness in knee joint ↑ *Mmp13, Col10, ADAMTS-5* gene expression in articular cartilages ↑ Igf1, IL-1β, Bmp2, Bmp4, Hif1α, Bax, Sox9, Vegf gene expression in knee joints ↑ FGF-23, FGFR-1 protein levels in knee joints↓ mineralization of hypertrophic chondrocytes (Meo Burt, Xiao et al. 2016) [89] ♂ ↓ *Sost, Dkk1, Lrp6* gene expression in knee joints ↑ *Wnt5a, Axin2, Lef1* gene expression in knee joints ↓ Sost, Lrp6 protein levels in knee joints ↑ Wnt7b, Wnt5a, Lrp5, Axin2, Gsk-3β, Lef1, nuclear β-catenin protein levels in knee joints ↑ *Mmp9* gene expression in femoral cartilage (Meo Burt, Xiao et al. 2018) [87] ♂/♀ ↑ signs of OA in knee joints (flattening of tibial plateau, osteophyte formation, sclerosis) ↓ femoral, tibial bone volume, trabecular number, thickness ↓ proteoglycane content, cartilage thickness in knee joints ↑ cartilage calcification in knee cartilage ♂ ↑ *Fgfr-1c, Fgf-18* gene expression in knee joints ↓ *Fgfr-3c* gene expression in knee joints ↑ FGF-2 protein level in subchondral bone ↑ MMP-13, SOX9, ADAMTS-5 protein level in articular cartilages ↓ Dkk1 Lrp6, Sost protein levels in articular cartilage (Xiao, Williams et al. 2020) [92]	♂ no radiographical signs of OA in knee joints (Meo Burt, Xiao et al. 2016) [89]

## 4. FGF-2 Isoforms in the Central Nervous System

The FGF family, especially FGF-2, is described as playing a crucial role in early and adult neurogenesis, although this effect has not yet been attributed to a specific FGF-2 isoform [96,97]. Apart from neurogenesis, FGF-2 also exerts multiple roles in neural cells that include cell proliferation, migration, synaptogenesis, survival and differentiation [18,98,99,100,101]. Due to these multifaceted neural effects, FGF-2 was shown to be involved in many neurologic and neurodegenerative diseases such as epilepsy, Huntington’s disease (HD), Parkinson’s disease (PD), Alzheimer’s disease (AD), multiple sclerosis (MS), spinal muscular atrophy (SMA), amyotrophic lateral sclerosis (ALS) as well as psychiatric disorders including anxiety, autism spectrum disorder (ASD), depression, schizophrenia and drug-related abusive disorders [22,102,103,104,105,106,107,108,109,110,111,112].

Today, comprehensive in vitro and in vivo studies as well as clinical trials exist evaluating the therapeutic administration (mostly FGF-2LMW) or overexpression of FGF-2 in the CNS. Furthermore, extensive investigations of FGF-2ko mice were conducted to evaluate loss of function, but without closer differentiation with regard to FGF-2 isoforms. So far, isoform-specific ko mice have only been employed to study dopaminergic (DA) nigral neurons and motor neurons (ALS mouse model) (Table 3) [22,113]. Regarding the nigrostriatal pathway, which is especially affected in PD, it was demonstrated that through maturation, mice lacking either FGF-2LMW or FGF-2HMW develop more DA neurons within the substantia nigra (SN) pars compacta, equally distributed to the hyperplasia phenotype previously described for adult mice lacking FGF-2 [99]. In FGF-2LMWko mice, this observation was traced back to an increased number of DA progenitor cells during the embryonic stage, followed by increased signs of regulatory apoptosis at the day of birth, even though it appeared that DA neurons were less affected. Interestingly, this was also true for FGF-2HMWko, despite being attenuated. Furthermore, it was confirmed that FGF-2LMW is the prominent FGF-2 isoform in wt mice during the embryonic stage, which was even more pronounced in FGF-2HMWko [21,22]. So far, compensatory upregulation of other FGFs or their receptors through development of the DA system has only been excluded for FGF-2ko mice [114], but was elusive for isoform-specific ko mice. With regard to that, supportive evidence for enhanced FGF-2LMW expression in FGF-2HMWko mice was also reported in the cardiovascular system [43,44]. Since FGF-2 immunoreactivity was reduced in remaining DA neurons of the SN in PD patients [115], its pathophysiological impact and neuroprotective potential were broadly investigated [29]. In the established neurotoxin-based 6-hydroxydopamine (6-OHDA) mouse model of PD, FGF-2ko mice were more severely affected than wt mice or mice overexpressing FGF-2 [100]. However, this observation has not yet been related to FGF-2LMW or FGF-2HMW. At least in vitro studies revealed that in addition to FGF-2LMW, also exogenously applied FGF-2HMW mediates survival-promoting activity on cultured embryonic ventral mesencephalic DA neurons [116]. Besides the DA system, an impaired cerebral cortex development was also reported in FGF-2ko mice as well as alterations in the hippocampal commissure and cervical spinal cord [18]. Furthermore, loss of FGF-2 was associated with hyperactivity and anxiety-like behavior in rodents due to imbalances in the glutamatergic and DA system as well as reduced hippocampal glucocorticoid receptors [117,118]. However, separated from the reported physical differences, no significant psychiatric alterations have been described in adult FGF-2 isoform-specific ko mice regarding their anxiety-like behavior, despite there being a slight tendency for reduced locomotor activity in male FGF-2LMWko mice, which, in contrast, was increased in FGF-2HMWko mice [22].

In research on ALS, the most common neurodegenerative motor neuron disease occurring in adulthood, FGF-2 isoform-specific ko mice were used to further elucidate the neuroprotective effect of complete or hemizygous ko of FGF-2 seen in mutant SOD1^G93A^ mice [104]. It has been previously hypothesized that excreted FGF-2 might act similarly to FGF-1, which readily binds to the FGFR-1 and leads to the spread of reactive astrocytosis, a hallmark of ALS implied in the non-cell autonomous motor neuron death mediated by astrocytes [119]. No beneficial effects were observed upon FGF-2LMW depletion. On the contrary, SOD1^G93A^FGF-2LMWko mice had a shorter life span, impaired motor performance and a poorer general condition compared to wt littermates. In addition, epidermal growth factor (EGF) mRNA expression was decreased in the gastrocnemius muscle of SOD1^G93A^FGF-2LMWko transgenic mice. In comparison, SOD1^G93A^FGF-2HMWko mice did not show any alterations, except that double mouse mutants with a heterozygous FGF-2HMWko showed a slight motor improvement in the rotarod test. Within the same study, co-culture of motor neurons with astrocytes devoid of either FGF-2LMW or FGF-2HMW (harvested from FGF-2LMWko or FGF-2HMWko mice) did not impact motor neuron growth, differentiation or gene expression patterns [113]. The fact that neither isoform influences motor neuron survival or differentiation in culture contradicts the involvement of the FGF-2 isoforms in the non-cell autonomous motor neuron death. One can, however, argue that embryonic motor neurons used in this study have not yet reached pathologic maturation and therefore did not show ALS-specific characteristics. Nonetheless, since FGF-2LMW depletion in vivo coincided with a decrease in muscular EGF expression in end-stage mice, an indirect co-regulation between the released FGF-2LMW and EGF in the neuron-muscle crosstalk is implied. With regard to gene expression, dysregulation of growth factors, specifically CNTF and GDNF, has also been described in the muscles and spinal cord of SOD1^G93A^FGF2ko mice as well as of FGF-2, CNTF and IGF-1 in SOD1^G93A^ mice with regular FGF-2 expression [104,120]. These results indicate disease stage-related differences in growth factor expression and interaction in ALS. Thus, loss of FGF-2 or either of its isoforms might trigger a compensatory co-regulation between growth factors regarding their gene transcription and translation.

During the last decade, not only in ALS but also in almost all neurodegenerative diseases and psychiatric disorders, there has been an evident shift towards pathologic innate immune response caused by exacerbated inflammation due to the dysfunction of macrophages, astrocytes comprising the blood brain barrier and microglia [121]. Liu et al. [122] demonstrated that FGF-2LMW regulates the retinoic acid inducible gene I (RIG-I) receptor’s innate immune response, as was proven when using mouse embryonic fibroblasts (MEFs) obtained from FGF-2LMWko mice. Moreover, RIG-I, an RNA helicase, which is known to function as a sensor for foreign viral RNA, was found to be stabilized by FGF-2LMW, as RIG-I protein levels were decreased when FGF-2LMW was lacking. Since mRNA levels remained unaltered, the authors hypothesize that FGF-2LMWko might impact the posttranslational modification of RIG-I. In addition, immune response titers of type I interferon, which include, interferon-α (IFN-α) and interferon-β (IFN-β), were determined after sendai virus (SeV) or polyinosinic-polycytidylic acid (polyIC) infection. Notably, both FGF-2LMWko MEFs and macrophages showed an increased mRNA expression of IFN-α and IFN-β as well as IL-6 and TNF-α following stimulation (Liu et al., 2015). The confirmed result was obtained for IFN-β while measuring the extracellular supernatant of FGF-2LMWko in both MEFs and macrophages. In agreement with that, also interferon regulatory factor 3 (IRF3) protein was activated as a result of SeV transduction in FGF-2LMWko MEFs. In combination, these aforementioned results provide first evidence of FGF-2LMW playing an important role in stabilizing RIG-I after infection and preventing its degradation by proteasomes. Moreover, FGF-2LMW acts as a negative regulator by maintaining the autoinhibition of RIG-I and thus, adjusting the level of RIG-I-mediated type I IFN during an innate immune response [122]. FGF-2 and RIG-I were also found to be involved in the miR-194 mediated viral response to the influenza A viral (IAV) infection. Specifically, it was shown that inhibition of miR-194 increases type I IFN production and therefore hinders viral replication and spread. Moreover, this IFN response was negatively regulated by miR-194 through FGF-2 and RIG-I suppression. Although within the study, it was not specified whether FGF-2HMW or FGF-2LMW was analyzed, the authors state that overexpression of FGF-2HMW in alveolar A549 cells enhanced type I IFN expression and suppressed viral spread during IAV infection [123]. This leads us to the conclusion that FGF-2 isoforms play a crucial role in regulating innate immune responses upon viral infection by regulating IFN levels through RIG-I dependent mechanisms.

Overall, the mentioned studies indicate subtle modifications mediated by FGF-2 isoforms in the CNS and innate immune system, even though more studies need to be performed in this field. With respect to the complexity of both systems, alterations might sometimes be masked and more apparent in total FGF-2ko mice. However, considering different disease pathophysiology as well as future therapeutic administration of either FGF-2LMW or FGF-2HMW in the CNS, its immune modulating effects should also be more precisely determined.

**Table 3 ijms-22-00390-t003:** In vivo alterations in the central nervous system of male (♂) and female (♀) FGF-2 isoform-specific knock out mice, lacking either FGF-2LMW (FGF-2LMWko) or FGF-2HMW (FGF-2HMWko) shown as increased (↑) or decreased (↓) compared to wildtype littermates. Additionally, in vitro data were provided, characterizing the participation of FGF-2LMW in the innate immune response by using FGF-2LMWko derived embryonic fibroblasts (MEF). ALS, amyotrophic lateral sclerosis; E14, embryonic day 14; DA, dopaminergic; EGF, epidermal growth factor; FGF-2, fibroblast growth factor 2; HMW, high molecular weight; IAV, influenza A virus; IL-6, interleukin 6; INF-α, interferon α; INF-β, interferon β; IRF3, interferon regulatory factor 3; LWW, low molecular weight; ko, knock out; MEF, mouse embryonic fibroblasts; P0, day of birth; polyIC, polynosinic-polycytidylic acid; RIG-1, retinoic acid inducible gene I; SeV, sendai virus; SNpc, substantia nigra pars compacta; SOD1G93A, superoxide dismutase 1 with a mutation on residue 93; TNF-α, tumor necrosis factor α.

Strain	FGF-2 LMWko (FGF-2^tm2Doe^)	FGF-2 HMWko (FGF-2^tm3Doe^)
Phenotype	♂/♀ no FGF-2HMW protein expression can be found in the embryonic nigrostriatal system ↑ number of DA precursor cells in the rostral subventricular zone at E14.5 ↑ signs of regulatory apoptosis in DA neurons of the ventral midbrain at P0 ↑ number of DA neurons in the adult female SNpc ↓ explorative behavior of adult male mice (von Hövel, Leiter et al. 2019) [22]	♂/♀ ↑ FGF-2LMW protein in brain tissue (Azhar, Yin et al. 2009) [43] ♂/♀ ↑ FGF-2LMW protein expression in the nigrostriatal pathway from P0 through development ↑ number of DA precursor cells in the rostral subventricular zone at E14.5 ↑ signs of apoptosis in DA neurons of the ventral midbrain at P0 ↑ number of DA neurons in the adult female SNpc ↑ explorative behavior of adult male mice (von Hövel, Leiter et al. 2019) [22]
ALS mouse model	♂/♀ SOD1G93A × FGF-2LMWko ↓ body weight, general condition score, and survival rate from 17 weeks onwards ↓ performance in the rotarod test in weeks 14–17 ↓ runtime in the footprint track with 21 weeks ↓ EGF mRNA expression in the gastrocnemius muscle at 21 weeks (Kefalakes, Sarikidi et al. 2019) [113]	♂/♀ SOD1G93A × FGF-2HMWko ↑ performance in the rotarod test in weeks 17, 18, 19, 20 and 21 (Kefalakes, Sarikidi et al. 2019) [113]
In vitro model	♂/♀ ↓ RIG-1 protein level in MEF independent of IAV infection ↑ INF-α, INF-β, IL-6, and TNF-α expression in macrophages and MEFs following polyIC or SeV stimulation ↑ phosphorylation of IRF3 following SeV infection (Liu, Luo et al. 2015) [122]	

## 5. Outlook

Since its first description, FGF-2 has been among the most studied growth factors in mammals that can be found in several organs and tissues [5,6]. However, until today, many of its physiological and pathophysiological effects still need to be clarified. Thereby, expression of distinct FGF-2 isoforms, summarized as FGF-2LMW and FGF-2HMW, which differ in their spatiotemporal expression pattern and mode of action, further enhance the complexity of the FGF-2 impact. Moreover, since FGF-2LMW is the homologous key sequence of all FGF-2 isoforms, no commercially available antibody can reliably distinguish between all isoforms, thus crucially limiting research options.

On this account, FGF-2 isoform-specific mice have been developed, including FGF-2LMWko mice and, vice versa, FGF-2HMWko mice. These strains are viable, fertile and reach a normal life span without macroscopic restrictions compared to wt mice [42,43,50]. Complementary, varying FGF-2 isoform-specific tg mice have been generated, comprising FGF-2LMWtg mice, overexpressing human or rat FGF-2LMW and FGF-2HMWtg mice, overexpressing either a single or a combination of human FGF-2HMW isoforms [45,46,47,48]. Similar to FGF-2 isoform-specific ko mice, these strains are also viable and fertile, whereby FGF-2HMWtg mice displayed osteomalacia and dwarfism together with age-related osteoporosis [45]. With respect to bone formation, further studies hypothesized a catabolic influence of FGF-2HMW in bone development, while FGF-2LMW functioned in an anabolic manner, which also benefited fracture healing. Apart from that, a cardioprotective potential was attributed to FGF2LMW in different heart disease models [48,55], suggesting a more valuable therapeutic potential for this isoform. In addition, through the development indicators of a compensatory upregulation of FGF-2LMW protein expression in FGF-2HMWko mice in the developing nigrostriatal system and heart exists [22,44]. However, since FGF-2HMW was shown to influence FGF-2 gene expression in vitro [36,37], this imbalance could also be the result of a dysregulation through loss of function rather than a compensatory upregulation, and should be defined in future studies. Indeed, a more detailed characterization regarding compensatory upregulation of other FGF family members or other growth factors in FGF-2 isoform-specific ko mice was elusive. For example, significant alterations of FGF-23 expression were registered in FGF-2HMWtg mice crucially influencing the manifestation of the visible bone phenotype through modulation of renal phosphate homeostasis [45]. Interestingly, FGF-2 was classified as a paracrinely acting growth factor in the past [124], which was verified by the present in vivo results for FGF-2LMW. In contrast, FGF-2HMW predominantly functions in an intracrine manner due to its nuclear restriction, enabling it to modulate other FGFs also resulting in profound endocrine effects. Besides that, sex-mediated endocrine effects should also be taken into account since it was demonstrated that E2 stimulates angiogenesis in FGF-2LMWko mice and also enhances FGF-2HMW expression [42].

To more precisely determine and localize the effects mediated by FGF-2 isoforms, their overexpression was driven by different promoters in FGF-2 isoform-specific tg mice. Thereby, the combination with the green fluorescent protein helped track the distribution of the FGF-2 isoforms and allowed the differentiation from endogenous FGF-2 [45,47]. Notably, overexpressed FGF-2 was already found in skeletal muscle and liver [47,48], but also other tissues could be conceivable, providing opportunities for future research of other organ systems using FGF-2LMWtg and FGF-2HMWtg mice. Overall, in future studies, sex matched groups should be analyzed to increase outcomes. Furthermore, the age range within groups should be minimized or determined precisely with regard to spatiotemporal expression patterns of FGF-2 isoforms. Of course, the conclusions achieved were of a higher quality whenever FGF-2 isoforms-specific ko mice and FGF-2 isoforms-specific tg mice were directly compared under similar conditions. Furthermore, identical genetic background should be ensured for all analyzed mice [22], or at least regarding the compared control group comprising the corresponding wt littermates.

Finally, the generated results of FGF-2 isoform-specific ko mice provide evidence for the physiological function of the missing isoform through development, normal function and disease. With respect to disease, the combination of established disease models and FGF-2 isoform-specific ko mice (independent of toxin administration, genetic engineering or crossbreeding) also provide indices of FGF-2LMW or FGF-2HMW mediated pathologic effects or their therapeutic potential. These results were complemented by characterizing FGF-2 isoform-specific tg mice, demonstrating alterations through development and maturation of different organ systems modulated by enhanced FGF-2LMW or FGF-2HMW expression. Ultimately, challenging FGF-2LMWtg and FGF-2HMWtg mice in established disease models could further evaluate the therapeutic potential of the different FGF-2 isoforms. 

## Figures and Tables

**Figure 1 ijms-22-00390-f001:**
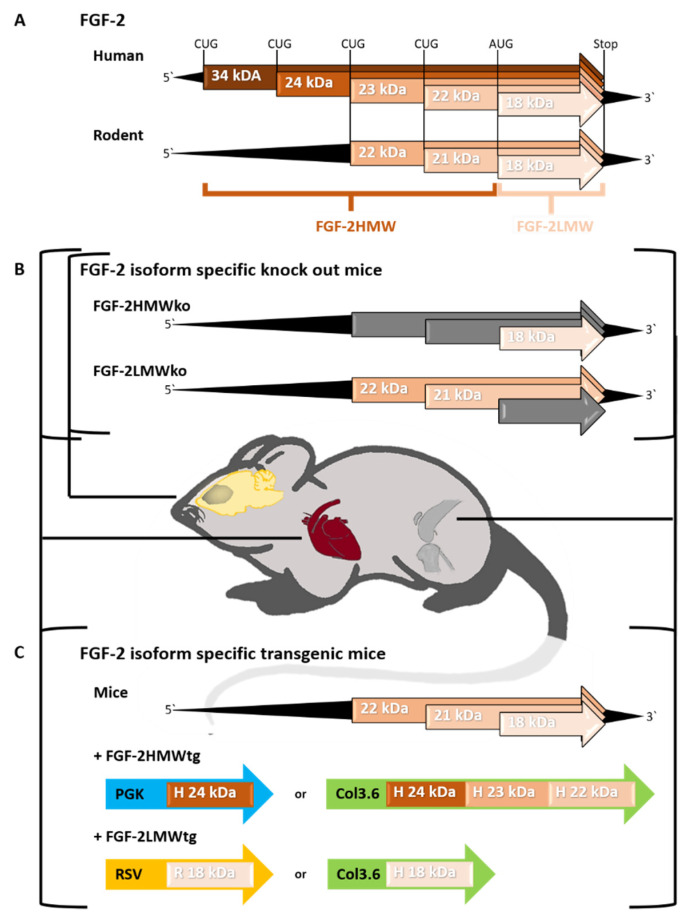
Endogenous FGF-2 expression and generation of FGF-2 isoform-specific mice. (**A**), based on one mRNA, translation from different CUG codons results in a 22, 23, 24, and 34 kDa product in humans summarized as FGF-2HMW. Meanwhile only the 21 and 22 kDa protein are present in rodents. Additionally, translation initiated at the AUG codon leads to expression of the 18 kDa FGF-2 isoform indicated as FGF-2LMW. (**B**), FGF-2 isoform-specific ko mice were created by gene targeting resulting in FGF-2HMWko mice, only expressing FGF-2LMW and vice versa FGF-2LMWko mice which only possess FGF-2HMW. These mice were more closely characterized with regard to development of the central nervous system and effects in neurodegenerative disease models as well as development and functionality of the cardiovascular system, bone formation and regeneration (indicated by square brackets). (**C**), Also FGF-2 isoform-specific tg mice were generated, which overexpress specific FGF-2 isoforms in addition to endogenous FGF-2. With regard to the cardiovascular system FGF-2HMWtg mice overexpressing the human 24 kDa FGF-2 isoform driven by the PGK were analyzed as well as FGF-2LMWtg mice overexpressing the rat FGF-2LMW protein using the RSV promoter. In opposition to that, bone physiology was characterized in FGF-2HMWtg mice additional possessing the human 22, 23, and 24 kDa FGF-2 isoform using the Col3.6 promoter, which also drives human FGF-2LMW isoform expression in FGF-2LMWtg mice. Col3.6, collagen 3.6 promoter; FGF-2, fibroblast growth factor 2; H, human; HMW, high molecular weight; kDA, kilo Dalton; ko, knock out; LWW, low molecular weight; PKC, protein kinase C; R, rat; RSV, Rous sarcoma virus long terminal repeat; tg, transgenic.

## Data Availability

Not applicable.

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
