# Peer review of "What Can We Learn from FGF-2 Isoform-Specific Mouse Mutants? Differential Insights into FGF-2 Physiology In Vivo"

_ijms, 2020, doi:10.3390/ijms22010390_

Round 1
Reviewer 1 Report
It is good to see that the authors have put in a lot of effort into extensively revising their manuscript - most importantly with the addition of data from FGF transgenic mouse studies.
Although I felt that the review could be made neater and leaner by perhaps moving some tables to the Supplementary files, the authors have not done this. Nevertheless, keeping these tables in the main manuscript now makes more sense as they are now more comprehensive and inclusive, with data from the FGF overexpression studies. Therefore, I support and applaud the new more comprehensive tables.
I am also glad to see that the authors have included a very helpful and clear diagram that sets out the field in an illustration in a nice way.
Author Response
Response to reviewer 1
Summary: It is good to see that the authors have put in a lot of effort into extensively revising their
manuscript - most importantly with the addition of data from FGF transgenic mouse studies.
Although I felt that the review could be made neater and leaner by perhaps moving some tables to the
Supplementary files, the authors have not done this. Nevertheless, keeping these tables in the main
manuscript now makes more sense as they are now more comprehensive and inclusive, with data from
the FGF overexpression studies. Therefore, I support and applaud the new more comprehensive tables.
I am also glad to see that the authors have included a very helpful and clear diagram that sets out the
field in an illustration in a nice way.
We are very grateful for the reviewer’s appreciation of our revised manuscript and his approval
towards the newly included tables and diagram, which we think highly increase comprehensibility
of our review
Reviewer 2 Report
The review appears to be a valuable collection of data regarding the effects of FGF-2 isoforms in several tissues, based on the in vivo studies. I believe that the manuscript is well written and efforts have been put to gather the information and present it synthetically in tables, which help the reader process the data easily. However, the manuscript would benefit of minor changes and of descriptive figures to illustrate FGF-2 roles in different tissues, as follows:
- Figure 1 is very useful for the reader and generally for understanding the main ideas in the review; what I suggest would be to move the information and references given in the figure legend to the text, I believe that this information given in the legend might be lost if it’s written only in the caption and that citation in the caption is somehow inappropriate.
- Section 2 dedicated to FGF-2 role in the cardiovascular system gathers very valuable information and thus this section deserves a separate figure in my opinion, synthesizing the main implications of FGF-2 in this system. The same for sections 3 and 4. I believe that this review would be highly improved if the Authors designed and included some explanatory figures for each sub-chapter- cardiovascular system, bone and central nervous system. The information is nicely gathered in tables, but the manuscript lacks figures.
- The Authors discuss the roles of FGF-2 isoforms in the functionality of the cardiovascular system, bone formation and regeneration as well as central nervous system, based on the information gained from in vivo studies. Still, I think it would be important to discuss in the introduction section the effects of FGF-2 in stem cells differentiation and development, as a starting point for the three types of tissues further discussed in the review.
- In my opinion, the title should be changed to a more clear and representative one for this paper.
Author Response
Response to reviewer 2
Summary: The review appears to be a valuable collection of data regarding the effects of FGF-2
isoforms in several tissues, based on the in vivo studies. I believe that the manuscript is well written and
efforts have been put to gather the information and present it synthetically in tables, which help the
reader process the data easily. However, the manuscript would benefit of minor changes and of
descriptive figures to illustrate FGF-2 roles in different tissues, as follows:
Comment No. 1)
Figure 1 is very useful for the reader and generally for understanding the main ideas in the review; what
I suggest would be to move the information and references given in the figure legend to the text, I
believe that this information given in the legend might be lost if it’s written only in the caption and that
citation in the caption is somehow inappropriate.
We are very pleased for the reviewer’s approval of the explanatory figure, which summarizes the
fundamental principles of the manuscript. In accordance to the reviewer’s suggestions the figure
legend was selectively adapted and citations remained exclusively in the main text.
Information for figure 1B can be found in line 78-82 (indicated in yellow since this paragraph was
not revised), meanwhile the caption was changed (indicated in green, line 114-126).
´ B, FGF-2 isoform-specific ko mice were created by gene targeting resulting in FGF-2HMWko mice,
only expressing FGF-2LMW and vice versa, FGF-2LMWko mice which only possess FGF-2HMW.
These mice were more closely characterized with regard to development of the central nervous
system and effects in neurodegenerative disease models as well as development and functionality
of the cardiovascular system, bone formation and regeneration (indicated by square brackets).´
In addition, the paragraph related to figure 1C in the main text (line 82-93, indicated in green) was
revised.
´Moreover, also different transgenic (tg) mice overexpressing either the FGF-2LMW (below
abbreviated to FGF-2LMWtg) or FGF-2HMW (below abbreviated to FGF-2HMWtg) isoform, in
addition to endogenous FGF-2, have been generated (Fig. 1C). Thereby, the applied promoters
differed with respect to the tissue of interest, to determine the physiological as well as
pathophysiological impact of FGF-2LMW and FGF-2HMW in vivo. In detail, created FGF-2HMWtg
mice either overexpress the human 24 kDa FGF-2 isoform driven by the protein kinase C (PGK)
promoter [1] or combined the human 22, 23, and 24 kDa FGF-2 isoform using the collagen 3.6
(Col3.6) promoter [2,3]. The latter promoter was also used to generate FGF-2LMWtg mice
overexpressing the human FGF-2LMW isoform as analyzed in detail with regard to bone
pathophysiology [3,4]. In contrast, FGF-2LMWtg mice overexpressing the rat FGF-2LMW protein
under the rous sarcoma virus long terminal repeat (RSV) promoter were applied in cardiovascular
research [5].´
And the caption of figure 1C was adapted in accordance (line 126-136, indicated in green).
`C, Also FGF-2 isoform-specific tg mice were generated, which overexpress specific FGF-2 isoforms
in addition to endogenous FGF-2. With regard to the cardiovascular system FGF-2HMWtg mice
overexpressing the human 24 kDa FGF-2 isoform driven by the PGK were analyzed as well as FGF2LMWtg mice overexpressing the rat FGF-2LMW protein using the RSV promoter. In opposition to
that, bone physiology was characterized in FGF-2HMWtg mice additional possessing the human 22,
23, and 24 kDa FGF-2 isoform using the Col3.6 promoter, which also drives human FGF-2LMW
isoform expression in FGF-2LMWtg mice.´
1. Liao, S.; Bodmer, J.R.; Azhar, M.; Newman, G.; Coffin, J.D.; Doetschman, T.; Schultz Jel, J. The influence
of FGF2 high molecular weight (HMW) isoforms in the development of cardiac ischemia-reperfusion
injury. J. Mol. Cell. Cardiol. 2010, 48, 1245-1254, doi:10.1016/j.yjmcc.2010.01.014.
2. Xiao, L.; Naganawa, T.; Lorenzo, J.; Carpenter, T.O.; Coffin, J.D.; Hurley, M.M. Nuclear isoforms of
fibroblast growth factor 2 are novel inducers of hypophosphatemia via modulation of FGF23 and
KLOTHO. J Biol Chem 2010, 285, 2834-2846, doi:10.1074/jbc.M109.030577.
3. Davis, M.G.; Zhou, M.; Ali, S.; Coffin, J.D.; Doetschman, T.; Dorn, G.W., 2nd. Intracrine and autocrine
effects of basic fibroblast growth factor in vascular smooth muscle cells. J. Mol. Cell. Cardiol. 1997, 29,
1061-1072, doi:10.1006/jmcc.1997.0383.
4. Xiao, L.; Liu, P.; Li, X.; Doetschman, T.; Coffin, J.D.; Drissi, H.; Hurley, M.M. Exported 18-kDa isoform of
fibroblast growth factor-2 is a critical determinant of bone mass in mice. J Biol Chem 2009, 284, 3170-
3182, doi:10.1074/jbc.M804900200.
5. Sheikh, F.; Sontag, D.P.; Fandrich, R.R.; Kardami, E.; Cattini, P.A. Overexpression of FGF-2 increases
cardiac myocyte viability after injury in isolated mouse hearts. Am J Physiol Heart Circ Physiol 2001,
280, H1039-1050, doi:10.1152/ajpheart.2001.280.3.H1039.
Comment No. 2)
Section 2 dedicated to FGF-2 role in the cardiovascular system gathers very valuable information and
thus this section deserves a separate figure in my opinion, synthesizing the main implications of FGF-2
in this system. The same for sections 3 and 4. I believe that this review would be highly improved if the
Authors designed and included some explanatory figures for each sub-chapter- cardiovascular system,
bone and central nervous system. The information is nicely gathered in tables, but the manuscript lacks
figures.
We thank the reviewer for the recommendation to further improve our manuscript by including
additional explanatory figures. However, according to our appreciation, the combination of figure 1
and the highly explanatory tables together with the text throughout provide a comprehensive
presentation. Besides the short time frame, we are convinced that additional figures would not
further increase the information in this case but just offer another mean of representation.
Comment No. 3)
The Authors discuss the roles of FGF-2 isoforms in the functionality of the cardiovascular system, bone
formation and regeneration as well as central nervous system, based on the information gained from
in vivo studies. Still, I think it would be important to discuss in the introduction section the effects of
FGF-2 in stem cells differentiation and development, as a starting point for the three types of tissues
further discussed in the review.
We agree with the reviewer suggestion to mention the role of FGF-2 in stem cell research in our
review. Thus, we have now included a short paragraph (line 41-48; indicated in green) regarding the
effects of FGF-2 in stem cell proliferation and differentiation.
`Furthermore, FGF-2 is widely used in stem cell research as an agent of self-renewal (proliferation)
and differentiation in vitro. Cultured stem cells first undergo a period of self-renewal during which
they maintain an undifferentiated homogenous state of pluripotency or multipotency. Addition of
different factors can act as stimuli of differentiation and thus, promote cell fate commitment to
lineages of interest [6-9]. Although many other signals are also involved in these processes, it is
implied that the concentration of FGF-2 plays a pivotal role regarding cell fate specification [10,11].´
Due to the extensive role of FGF-2 in stem cell research we decided to keep this paragraph in the
introduction as recommended, rather than describe its effects on the three tissues of interest
(cardiovascular system, bone formation and regeneration as well as neural development and
disease) in detail. In this way we tried to avoid shifting the focus on in vitro studies which is out of
scope of this review as its main goal is to provide an overview of isoform-specific findings in vivo.
6. Amit, M.; Carpenter, M.K.; Inokuma, M.S.; Chiu, C.P.; Harris, C.P.; Waknitz, M.A.; Itskovitz-Eldor, J.;
Thomson, J.A. Clonally derived human embryonic stem cell lines maintain pluripotency and
proliferative potential for prolonged periods of culture. Dev. Biol. 2000, 227, 271-278,
doi:10.1006/dbio.2000.9912.
7. Kawai, T.; Takahashi, T.; Esaki, M.; Ushikoshi, H.; Nagano, S.; Fujiwara, H.; Kosai, K. Efficient
cardiomyogenic differentiation of embryonic stem cell by fibroblast growth factor 2 and bone
morphogenetic protein 2. Circ. J. 2004, 68, 691-702, doi:10.1253/circj.68.691.
8. Kizhner, T.; Ben-David, D.; Rom, E.; Yayon, A.; Livne, E. Effects of FGF2 and FGF9 on osteogenic
differentiation of bone marrow-derived progenitors. In Vitro Cell. Dev. Biol. Anim. 2011, 47, 294-301,
doi:10.1007/s11626-011-9390-y.
9. Lotz, S.; Goderie, S.; Tokas, N.; Hirsch, S.E.; Ahmad, F.; Corneo, B.; Le, S.; Banerjee, A.; Kane, R.S.; Stern,
J.H., et al. Sustained levels of FGF2 maintain undifferentiated stem cell cultures with biweekly feeding.
PLoS One 2013, 8, e56289, doi:10.1371/journal.pone.0056289.
10. Qian, X.; Davis, A.A.; Goderie, S.K.; Temple, S. FGF2 concentration regulates the generation of neurons
and glia from multipotent cortical stem cells. Neuron 1997, 18, 81-93, doi:10.1016/s0896-
6273(01)80048-9.
11. Mimura, S.; Suga, M.; Liu, Y.; Kinehara, M.; Yanagihara, K.; Ohnuma, K.; Nikawa, H.; Furue, M.K.
Synergistic effects of FGF-2 and Activin A on early neural differentiation of human pluripotent stem
cells. In Vitro Cell. Dev. Biol. Anim. 2015, 51, 769-775, doi:10.1007/s11626-015-9909-8.
Comment No. 4)
In my opinion, the title should be changed to a more clear and representative one for this paper.
We are thankful for the reviewer’s recommendation; however, we ensure the title of our review
was chosen with careful consideration. As already stated above we aimed to provide a unique
overview of the in vivo results of FGF-2 isoform-specific mouse mutants as the basis for future
research. On this account, we are convinced that our aim as well as content of the manuscript is
perfectly summarized and expressed by the chosen title ‘What can we learn from FGF-2 isoformspecific mouse mutants? Differential insights into FGF-2 physiology in vivo’.